# The Cloud Feedback Model Intercomparison Project Observational Simulator Package: Version 2

Dustin J. Swales[1,2], Robert Pincus[1,2], Alejandro Bodas-Salcedo[3]

[1] Cooperative Institute for Research in Environmental Sciences, University of Colorado Boulder, Boulder, Colorado, US
[2] NOAA/Earth System Research Laboratory, Boulder, Colorado, US
[3] Met Office Hadley Centre, Exeter, UK

*Correspondence to*: Dustin Swales (dustin.swales@noaa.gov)

**Abstract.** The Cloud Feedback Model Intercomparison Project Observational Simulator Package (COSP) gathers together a collection of observation proxies or "satellite simulators" that translate model-simulated cloud properties to synthetic observations as would be obtained by a range of satellite observing systems. This paper introduces COSP 2, an evolution focusing on more explicit and consistent separation between host model, coupling infrastructure, and individual observing proxies. Revisions also enhance flexibility by allowing for model-specific representation of sub-grid scale cloudiness, provide greater clarity by clearly separating tasks, support greater use of shared code and data including shared inputs across simulators, and follow more uniform software standards to simplify implementation across a wide range of platforms. The complete package including a testing suite is freely available.

## 1 A common language for clouds

The most recent revision to the protocols for the Coupled Model Intercomparision Project (CMIP, see Eyring et al., 2016) includes a set of four experiments for the Diagnosis, Evaluation, and Characterization of Klima (Climate). As the name implies one intent of these experiments is to evaluate model fields against observations, especially in simulations in which sea-surface temperatures are prescribed to follow historical observations. Such an evaluation is particularly important for clouds since these are a primary control on the Earth's radiation budget.

But such a comparison is not straightforward. The most comprehensive views of clouds are provided by satellite remote sensing observations. Comparisons to these observations are hampered by the large discrepancy between the model representation, as profiles of bulk macro- and microphysical cloud properties, and the information available in the observations which may, for example, be sensitive only to column-integrated properties or be subject to sampling issues caused by limited measurement sensitivity or signal attenuation. To make comparisons more robust the Cloud Feedback

Model Intercomparison Project (CFMIP, https://www.earthsystemcog.org/projects/cfmip/) has led efforts to apply observation proxies or "instrument simulators" to climate model simulations made in support of the (CMIP) and CFMIP.

Instrument simulators are diagnostic tools that map the model state into synthetic observations. The ISCCP (International Satellite Cloud Climatology Project) simulator (Klein and Jakob, 1999; Webb et al., 2001), for example, maps a specific representation of cloudiness to aggregated estimates of cloud-top pressure and optical thickness as would be provided by a particular satellite observing program, accounting for sampling artifacts such as the masking of high clouds by low clouds and providing statistical summaries computed in precise analogy to the observational datasets. Subsequent efforts have produced simulators for other passive instruments include MISR (the Multi-angle Imaging SpectroRadiometer: Marchand and Ackerman, 2010 describe this simulator) and MODIS (Moderate Resolution Imaging Spectroradiometer; Pincus et al., 2012) and for the active platforms CALIPSO (Cloud-Aerosol Lidar and Infrared Pathfinder Satellite Observation; Chepfer et al., 2008) and CloudSat (Haynes et al., 2007).

Some climate models participating in the initial phase of CFMIP provided results from the ISCCP simulator. To ease the way for adoption of multiple simulators CFMIP organized the development of the Observation Simulator Package (COSP; Bodas-Salcedo et al., 2011). A complete list of the instrument simulator diagnostics available in COSP1 and COSP2, can be found in Bodas-Salcedo et al., 2011. The initial implementation, hereafter COSP1, supported more widespread and thorough diagnostic output requested as part of the second phase of CFMIP associated with CMIP5 (Taylor et al., 2012). Similar but somewhat broader requests are made as part of CFMIP3 (Webb et al., 2017) and CMIP6 (Eyring et al., 2016).

The view of model clouds enabled by COSP has enabled important advances. Results from COSP have been useful in identifying biases in the distribution of model-simulated clouds within individual models (Kay et al., 2012; Nam and Quaas, 2012), across the collection of models participating in coordinated experiments (Nam et al., 2012), and across model generations (Klein et al., 2013). Combined results from active and passive sensors have highlighted tensions between process fidelity and the ability of models to reproduce historical warming (Suzuki et al., 2013), while synthetic observations from the CALIPSO simulator have demonstrated how changes in vertical structure may provide the most robust measure of climate change on clouds (Chepfer et al., 2014). Results from the ISCCP simulator have been used to estimate cloud feedbacks and adjustments (Zelinka et al., 2013) through the use of radiative kernels (Zelinka et al., 2012).

COSP1 simplified the implementation of multiple simulators within climate models but treated many components, especially the underlying simulators contributed by a range of collaborators, as inviolate. After most of a decade this approach was showing its age, as we detail in the next section. Section 3 describes details the conceptual model underlying a new implementation of COSP and a design that addresses these issues. Section 4 provides some details regarding implementation. Section 5 contains a summary of COSP2 and provides information about obtaining and building the software.

## 2 Barriers to consistency, efficiency, and extensibility

Especially in the context of cloud feedbacks, diagnostic information about clouds is most helpful when it is consistent with the radiative fluxes with which the model state co-evolves. COSP2 primarily seeks to address a range of difficulties that arose in maintaining this consistency in COSP1 as the package became used in an increasingly wide range of models. For example, as COSP1 was implemented in a handful of models, it became clear that differing cloud microphysics across models would often require substantial code changes to maintain consistency between COSP1 and the host model.

The satellite observations COSP emulates are derived from individual observations made on spatial scales of order kilometres (for active sensors, tens of meters) and statistically summarized at ~100 km scales commensurate with model predictions and aggregated observational data streams. To represent this scale-bridging the ISSCP simulator introduced the idea of subcolumns – discrete, homogenous samples constructed so that a large ensemble reproduces the profile of bulk cloud properties within a model grid column and any overlap assumptions made about vertical structure. COSP1 inherited the specific methods for generating subcolumns from the ISCCP simulator including a fixed set of inputs (convective and stratiform cloud fractions, visible-wavelength optical thickness for ice and liquid, mid-infrared emissivity) describing the distribution of cloudiness. Host models for which this description wasn't appropriate, for example a model in which more than one category of ice was considered in the radiation calculation (Kay et al., 2012), had to make extensive changes to COSP if the diagnostics were to be informative.

The fixed set of inputs limited models' ability to remain consistent with the radiation calculations. Many global models now use the Monte Carlo Independent Column Approximation (Pincus et al., 2003) to represent subgrid-scale cloud variability in radiation calculations. Inspired by the ISCCP simulator, McICA randomly assigns subcolumns to spectral intervals, replacing a two-dimensional integral over cloud state and wavelength with a Monte Carlo sample. Models using McICA for radiation calculations must implement methods for generating subcolumns, and the inability to share these calculations between radiation and diagnostic calculations was neither efficient nor self-consistent.

COSP1 was effective in packaging together a set of simulators developed independently and without coordination but this had its costs. COSP1 contains three independent routines for computing joint histograms, for example. Simulators required inputs, some closely related (relative and specific humidity, for example) and produced arbitrary mixes of outputs at the column and subcolumn scale, making multi-sensor analyses difficult.

## 3 A conceptual model and the resulting design

Though the division was not always apparent in COSP1, all satellite simulators perform four discrete tasks within each column:

1. Sampling of cloud properties to create homogenous subcolumns
2. Mapping of cloud physical properties (e.g. condensate concentrations and particle sizes) to relevant optical properties (optical depth, single scattering albedo, radar reflectivity, etc.)
3. Synthetic retrievals of individual observations (e.g. profiles of attenuated lidar backscatter or cloud-top pressure/column optical thickness pairs)
4. Statistical summarization (e.g. appropriate averaging or computation of histograms)

The first two steps require detailed knowledge as to how a host model represents cloud physical properties; the last two steps mimic the observational process. This first step is not invoked for models with high spatial resolution, as we describe more fully below.

The design of COSP2 reflects this conceptual model. The primary inputs to COSP2 are subcolumns of optical properties (i.e. the result of step 2 above), and it is the host model's responsibility to generate subcolumns and map physical to optical properties consistent with model formulation. This choice allows models to leverage infrastructure for radiation codes using McICA, making radiation and diagnostic calculations consistent with one another. COSP2 also requires as input a small set of column-scale quantities including surface properties and thermodynamic profiles. These are used, for example, by the ISCCP simulator to mimic the retrieval of cloud-top pressure from infrared brightness temperature. In COSP 2 the instrument simulator components have no dependencies on the host model including the underlying spatial scale.

Simulators within COSP2 are explicitly divided into two components (Figure 1). The subcolumn simulators, shown as lenses with colors representing the sensor being mimicked, take a range of column inputs (ovals) and subcolumn inputs (circles, with stacks representing multiple samples) and produce synthetic retrievals on the subcolumn scale, shown as stacks of squares. Column simulators, drawn as funnels, reduce these subcolumn synthetic retrievals to statistical summaries (hexagons). Column simulators may summarize information from a single observing system, as indicated by shared colors. Other column simulators may synthesize subcolumn retrievals from multiple sources, as suggested by the black funnel.

This division mirrors the processing of satellite observations by space agencies. At NASA, for example, these processing steps correspond to the production of Level 2 and Level 3 data, respectively. Implementation required the restructuring of many of the component simulators from COSP1. This allowed for modest code simplification by using common routines to make statistical calculations.

Models with spatial resolution roughly commensurate with individual satellite observations might apply the subcolumn simulators directly to model columns, then report statistics at some reduced spatial resolution. The scale separation is also illustrated by COSP implementations in multi-scale modeling frameworks (e.g. Marchand and Ackerman, 2010), in which

the subcolumn simulators are applied to individual high-resolution cloud-scale columns and statistical summaries are reported on the low-resolution global grid.

Separating the computation of optical properties from the description of individual simulators allows for modestly increased
efficiency because inputs shared across simulators, for example the 0.67 µm optical depth required by the ISCCP, MODIS, and MISR simulators, do not need to be recomputed or copied. The division also allowed us to make some simulators more generic. In particular, the CloudSat simulator used by COSP is based on the Quickbeam package (Haynes et al., 2007). Quickbeam is quite generic with respect to radar frequency and the location of a sensor but this flexibility was lost in COSP1. COSP2 exposes the generic nature of the underlying subcolumn lidar and radar simulators and introduces
configuration variables that provide instrument-specific information to the subcolumn calculation.

## 4 Implementation

### 4.1 Interface and control flow

The simplest call to COSP now makes use of three Fortran derived types representing the column and subcolumn inputs and the desired outputs. The components of these types are `PUBLIC` (that is, accessible by user code) and are, with few
exceptions, pointers to appropriately-dimensioned arrays. COSP determines which subcolumn and column simulators are to be run based on the allocation status of these arrays, as described below. All required subcolumn simulators are invoked, followed by all column simulators. Optional arguments can be provided to restrict work to a subset of the provided domain (set of columns) to limit memory use.

COSP2 has no explicit way of controlling which simulators are to be invoked. Instead, column simulators are invoked if space for one or more outputs is allocated – that is, if one or more of output variables (themselves components of the output derived type) is associated with array memory of the correct shape. The set of column simulators determines which subcolumn simulators are to be run. Not providing the inputs to these subcolumn simulators is an error.

The use of derived types allows COSP's capabilities to be expanded incrementally. Adding a new simulator, for example, requires adding new components to the derived type representing inputs and outputs but codes referring to existing components of those types need not be changed. This functionality is already in use – the output fields available in COSP2 extend COSP1's capabilities to include the joint histograms of optical thickness and effective radius requested as part of CFMIP 3.

**4.2 Enhancing portability**

COSP2 also includes a range of changes aimed at providing more robust, portable, and/or flexible code, many of which were suggested by one or more modeling centers using COSP. These include

1. Robust error checking, implemented as a single routine which validates array shapes and physical bounds on values.
2. Error reporting standardized to return strings, where non-null values indicates failure.
3. Parameterized precision for all `REAL` variables (`KIND=wp`) where the value of `wp` can be set in a single location to correspond to 32 or 64 byte real values.
4. Explicit `INTENT` for all subroutine arguments.
5. Standardization of vertical ordering for arrays in which the top of the domain is index 1.
6. Conformance with Fortran 2003 standards.

COSP2 must also be explicitly initialized before use. The initialization routine calls routines for each simulator in turn. This allows for more flexible updating of ancillary data such as lookup tables.

**5. Summary**

Version 2 of the CFMIP Observational Simulator Package, COSP2, represents a substantial revision of the COSP platform. The primary goal was to allow a more flexible and representation of clouds, so that the diagnostics produced by COSP can be fully consistent with radiation calculations made by the host model, even in the face of increasingly complex descriptions of cloud macro- and microphysical properties. Consistency requires that host models generate subcolumns and compute optical properties, so that the interface to the host model is entirely revised relative to COSP1.

As an example and a bridge to past efforts, COSP2 includes an optional layer that provides compatibility with COSP 1.4.1, accepting the same inputs and implementing sampling and optical property calculations in the same way. COSP2, either via this COSP 1.4.1. interface or via mode direct implementations, may be used to provide CMIP6/CFMIP3 output.

Simulators in COSP2 are divided into those that compute subcolumn (pixel) scale synthetic retrievals and those that compute column (grid) scale statistical summaries. This distinction, and the use of extensible derived types in the interface to the host model, are designed to make it easier to extend COSP's capabilities by adding new simulators at either scale, including analysis making use of observations from multiple sources.

**6. Code availability**

The source code for COSP2, along with downloading and installation instructions, are available in a GitHub repository (https://github.com/CFMIP/COSPv2.0). This manuscript is based on commit 04df31a, which is also available at https://doi.org/10.5281/zenodo.1040332. Previous versions of COSP (e.g. v1.3.1, v1.3.2, v1.4.0 and v1.4.1) are available in a

parallel repository (https://github.com/CFMIP/COSPv1). But these versions have reached the end of life, and COSP2 provides the basis for future development. Models updating or implementing COSP, or developers wishing to add new capabilities, are best served by starting with COSP2.

**Acknowledgements**

The authors thank the COSP Project Management Committee for guidance and Tomoo Ogura for testing the implementation

of COSP2 in the MIROC climate model. Dustin Swales and Robert Pincus were financially supported by NASA under award NNX14AF17G. A. Bodas-Salcedo received funding from the IS-ENES2 project, European FP7-INFRASTRUCTURES-2012-1 call (Grant Agreement 312979).

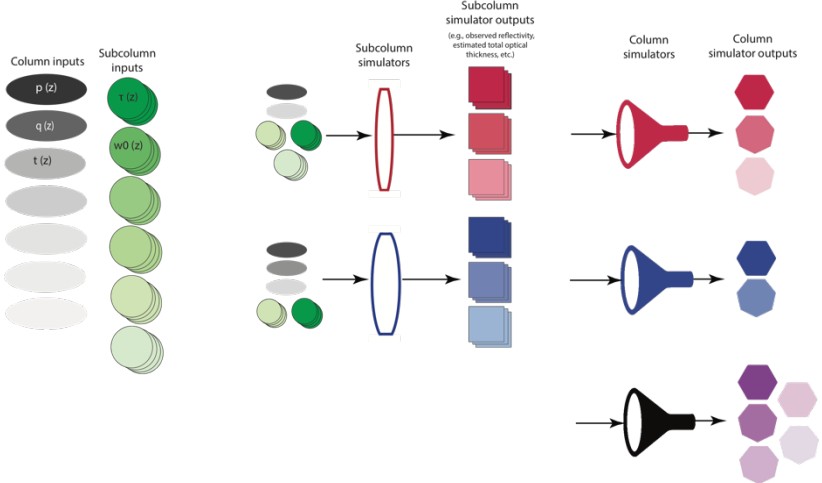

**Figure 1. Organizational view of COSP2. Within each grid cell host models provide a range physical inputs at the grid scale (grey ovals, one profile per variable) and optical properties at the cloud scale (green circles, Nsubcol profiles per variable). Individual subcolumn simulators (lens shapes, colored to indicate simulator types) produce Nsubcol synthetic retrievals (squares) which are then summarized by aggregation routines (funnel shapes) taking input from one or more subcolum simulators.**

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
