# Peer review of "The Cloud Feedback Model Intercomparison Project Observational Simulator Package: Version 2"

_Geoscientific Model Development, 2017_

## Short Comment (SC1) · 9 Aug 2017

Dear authors,

I tried to access the sources on github and noticed the links given in the manuscript (p.5, ll. 21-22) are not correct. The right locations are: https://github.com/CFMIP/COSPv2.0 and https://github.com/CFMIP/COSPv1, respectively.

Kind regards, Bastian

---

## Short Comment (SC2) · 9 Aug 2017

Dear Bastian,

Thanks for bringing this to our attention. I will correct this before publication.

Best regards, Dustin Swales

---

## Referee Comment (RC1) · Anonymous Referee #1 · 13 Sep 2017

This is an exceptionally well-written manuscript which introduces a new version of the CFMIP Observational Simulator Package (COSP). The manuscript clearly described the design of COSPv2 and its software improvements compared to the COSPv1. The reorganization of the COSP architecture allows for increased efficiency, helps to make the diagnostics consistent with radiation calculations of the host model more easily, and makes it easier to add new simulators and diagnostics. Given the wide use of the COSP in the global climate modeling community, this article should be able to provide helpful guidance to users.

Specific comments: 1. It will be better if the author can quantitatively estimate the

improved efficiency of the new COSP version compared with the old one in section 3. 2. Since COSPv1.4.1 is the production version for CFMIP3 and CMIP6, will the new COSPv2 diagnostics be different? It was also mentioned in the summary that there is an optional layer in COSPv2 to provide compatibility with COSPv1.4.1. Is this option recommended for recent efforts of model evaluation? 3. Page 3, Line 10, "ISSCP" should be "ISCCP".

---

## Referee Comment (RC2) · B. Kern (Referee) · 14 Sep 2017

**General comments**

The manuscript describes version 2 of the CFMIP Observational Simulator Package (COSP). Especially enhancements in the software structure to disentangle the diagnostic modules, the coupling interface and the host model.

The manuscript is well written and easy to follow. The developments of the software to enhance modularisation is appreciated and should facilitate integration of the diagnostics in numerical models, as well as the integration of novel diagnostic modules in

COSP itself. As a technical paper, describing developments of a novel version of the COSP, it fits in the scope of the journal and should be published, subject to few minor comments.

Focusing on the novel interface is a good choice and keeps the manuscript at reasonable length. I assume measurements of computational demands vary over a wide range, depending on the complexity of the simulator package, and thus would not be very beneficial. Details of COSP and on the simulator modules can be found in a previous paper on version 1, this may be stressed a bit more (yes, I know it is cited on p.2 l.14).

There are several acronyms of satellite platforms and sensors (especially p.2 ll.4ff.). All the references are given and the acronyms are well known (at least in parts of the community), but maybe you could include the acronyms "decryption" (in-line, table, or list of acronyms?).

**Specific comments**

I have only one specific comment, the second part is more a suggestion on how to support developers integrating the COSP in their numerical models (and is a bit beyond the publication of the paper).

On p.4 l.10ff:
It seems clear to me, that for a coarse resolution general circulation model, one has to sample some kind of subcolumns, to reach a horizontal resolution compatible with the simulator modules. What, if using a high resolution model (1km or smaller)? Can columns be passed directly and "column-scale" properties have to be aggregated to a resolution suitable for the simulators (ISCCP)? Of course, you write, *"it is the host model's responsibility to generate subcolumns and map physical to optical properties consistent with model formulation"*. So, it should be the responsibility of the developer integrating the interface in a numerical model to provide the proper input fields, but

maybe you could add some hints on that.

It may be beneficial to have more details on the interface routines and the in- and output fields, which have to be used in the host model. If you do not want to bore the reader with too technical description, maybe you could think about a user's manual in the repository or as a supplement to the paper.

That leads me to an additional comment, which is not crucial for publication of the paper:

I also retrieved the code from github and managed to compile it and run the provided test routines. This was more or less straightforward (it took me some time, because I had to compile CMOR2 first).

However, there are some minor inconsistencies in the README(.txt) files (some changed filenames, *cosp_interface_v1p5.f90* mentioned in README not available). It is very good, that you include examples and testing routines in the repository. With the README files and the code examples, I think, I might be able to include the interface in a numerical model. For me it is fine to have the documentation in the README files and in the code. But maybe it would be more convenient to have an overview of the interface routines and details of in- and output fields in one place. So, you may think about a small user's manual as pdf in the repository or as supplement to the paper (there seems to be one for COSP 1.3.1) also including more technical details on the interface routines. It might ease the integration of COSP in numerical models.

**Technical corrections**

p.1, l.20:
Please include the acronym *CMIP* here, as it is used later in the text.

p.2, l.16:
Please update the reference *Webb et al., 2016* to *Webb et al., 2017* (see also below)

p.6, l.19:

Please include the section: *Code availability*
cf. https://www.geoscientific-model-development.net/about/manuscript_types.html
*In the case where new code is described in the paper, this is subject to the same availability requirements as for complete model descriptions. The code should be made available, and a model availability paragraph must be included.*

p.7, l.18:
Please change *Geosci. Model Dev. Disc.* to *Geosci. Model Dev.*

p.8, ll.20ff.:
The final revised version of this article is published:
*Webb, M. J., Andrews, T., Bodas-Salcedo, A., Bony, S., Bretherton, C. S., Chadwick, R., Chepfer, H., Douville, H., Good, P., Kay, J. E., Klein, S. A., Marchand, R., Medeiros, B., Siebesma, A. P., Skinner, C. B., Stevens, B., Tselioudis, G., Tsushima, Y., and Watanabe, M.: The Cloud Feedback Model Intercomparison Project (CFMIP) contribution to CMIP6, Geosci. Model Dev., 10, 359-384, doi:10.5194/gmd-10-359-2017, 2017.*

---

## Short Comment (SC3) · 11 Oct 2017

Dustin

GMD is encouraging authors to provide a persistent access to the released source code through the use of a DOI which then can be cited in the paper. For projects in GitHub a DOI can be created using Zenodo, see https://guides.github.com/activities/citable-code/ for details. Please note that in the code accessibility section you can still point the reader to the GitHub repository for the newest version even if you use a DOI for the relevant release.

All the best wishes Lutz Gross GMD Executive Editor

---

## Author Comment (AC1) · 6 Nov 2017

Dear Lutz,

Thank you for this. I've created a DOI and will include it in the final version of the manuscript.

Best regards, Dustin Swales

---

## Author Comment (AC2)

The authors would like to thank the anonymous referee for providing comments on this manuscript. Our responses are in blue, just below the referee comments.

This is an exceptionally well-written manuscript which introduces a new version of the CFMIP Observational Simulator Package (COSP). The manuscript clearly described the design of COSPv2 and its software improvements compared to the COSPv1. The reorganization of the COSP architecture allows for increased efficiency, helps to make the diagnostics consistent with radiation calculations of the host model more easily, and makes it easier to add new simulators and diagnostics. Given the wide use of the COSP in the global climate modeling community, this article should be able to provide helpful guidance to users.

*Comment 1: It will be better if the author can quantitatively estimate the improved efficiency of the new COSP version compared with the old one in section 3.*

In the text, we refer to "modest" increases in performance as a result of removing memory copies and redundant calculations in COSP2. Unfortunately, it's not feasible to compare COSP1 and COSP2 timing results on such a granular level, since the codes are organized very differently. With that being said, we're confident to say that computing a field once instead of three times is computationally more efficient.

From our experiences running COSP2 inline with a GCM (CAM), we observe roughly a ~65% speedup in COSP2 runtime when compared to COSP1. However, since we only tested this implementation in one model, we are reluctant to say that this performance increase is robust across a range of architectures and testing COSP2 across a range of models is beyond the scope of this work.

Comment 2a: Since COSPv1.4.1 is the production version for CFMIP3 and CMIP6, will the new COSPv2 diagnostics be different?

The diagnostics from COSPv1.4.1 are scientifically equivalent to the diagnostics produced by COSPv2.

Comment 2b: It was also mentioned in the summary that there is an optional layer in COSPv2 to provide compatibility with COSPv1.4.1. Is this option recommended for recent efforts of model evaluation?

Provided with COSP2 is an interface designed to be a "drop-in" replacement for COSP1.4.1. This is intended for modeling centers to implement COSP2 in their models without having to make code modifications. However, if you are new to using COSP for model validation/evaluation, we suggest starting directly with COSP2, as the 1.4.1 interface is more or less intended for legacy COSP1 users to use as a "bridge" between COSP1 and COSP2.

Comment 3: Page 3, Line 10, "ISSCP" should be "ISCCP". Changed in manuscript.

---

## Author Comment (AC3)

The authors would like to thank Bastian Kern for providing comments on this manuscript. Our responses are in blue, just below the referee comments.

The manuscript describes version 2 of the CFMIP Observational Simulator Package (COSP). Especially enhancements in the software structure to disentangle the diag-nostic modules, the coupling interface and the host model.

The manuscript is well written and easy to follow. The developments of the software to enhance modularisation is appreciated and should facilitate integration of the diagnostics in numerical models, as well as the integration of novel diagnostic modules in COSP itself. As a technical paper, describing developments of a novel version of the COSP, it fits in the scope of the journal and should be published, subject to few minor comments.

Focusing on the novel interface is a good choice and keeps the manuscript at reasonable length. I assume measurements of computational demands vary over a wide range, depending on the complexity of the simulator package, and thus would not be very beneficial. Details of COSP and on the simulator modules can be found in a previous paper on version 1, this may be stressed a bit more (yes, I know it is cited on p.2 I.14).

We added a sentence into the text guiding readers to the COSP1 paper for more information on the diagnostics available in COSP1/COSP2.

There are several acronyms of satellite platforms and sensors (especially p.2 ll.4ff.). All the references are given and the acronyms are well known (at least in parts of the community), but maybe you could include the acronyms "decryption" (in-line, table, or list of acronyms?).

Very good point. In the text (see p.2. l.5-14) we added the acronym definitions for the various instruments.

**Specific comments**

I have only one specific comment, the second part is more a suggestion on how to support developers integrating the COSP in their numerical models (and is a bit beyond the publication of the paper).

**On p.4 l.10ff:**

It seems clear to me, that for a coarse resolution general circulation model, one has to sample some kind of subcolumns, to reach a horizontal resolution compatible with the simulator modules. What, if using a high resolution model (1km or smaller)? Can columns be passed directly and "column-scale" properties have to be aggregated to a resolution suitable for the simulators (ISCCP)? Of course, you write, "it is the host model's responsibility to generate subcolumns and map physical to optical properties consistent with model formulation". So, it should be the responsibility of the developer integrating the interface in a numerical model to provide the proper input fields, but maybe you could add some hints on that.

Just as in previous versions, when using a high-resolution model, model-columns can (and should) be passed directly to into COSP. This was something we did not stress in the text,

but should have, as it's in the COSP1 paper. We added a few sentences (see p.4 l.31) in the text explaining this.

It may be beneficial to have more details on the interface routines and the in- and output fields, which have to be used in the host model. If you do not want to bore the reader with too technical description, maybe you could think about a user's manual in the repository or as a supplement to the paper.

That leads me to an additional comment, which is not crucial for publication of the pa-per: I also retrieved the code from github and managed to compile it and run the provided test routines. This was more or less straightforward (it took me some time, because I had to compile CMOR2 first).

However, there are some minor inconsistencies in the README(.txt) files (some changed filenames, cosp\_interface\_v1p5.f90 mentioned in README not available).

We've updated all of the README files throughout COSP.

It is very good, that you include examples and testing routines in the repository. With the README files and the code examples, I think, I might be able to include the interface in a numerical model. For me it is fine to have the documentation in the README files and in the code. But maybe it would be more convenient to have an overview of the interface routines and details of in- and output fields in one place. So, you may think about a small user's manual as pdf in the repository or as supplement to the paper (there seems to be one for COSP 1.3.1) also including more technical details on the interface routines. It might ease the integration of COSP in numerical models.

**Technical corrections**

p.1, l.20:

Please include the acronym CMIP here, as it is used later in the text.

Corrected in text.

p.2, l.16:

Please update the reference Webb et al., 2016 to Webb et al., 2017 (see also below) p.6, l.19: Corrected in text.

Please include the section: Code availability

Added new section to text. Previously the code was described in the summary section and not in its own section.

p.7, l.18:

Please change Geosci. Model Dev. Disc. to Geosci. Model Dev.

Corrected in text.

p.8, II.20ff.:

The final revised version of this article is published:

Webb, M. J., Andrews, T., Bodas-Salcedo, A., Bony, S., Bretherton, C. S., Chadwick, R., Chepfer, H., Douville, H., Good, P., Kay, J. E., Klein, S. A., Marchand, R., Medeiros, B., Siebesma, A. P., Skinner, C. B., Stevens, B., Tselioudis, G., Tsushima, Y., and Watanabe, M.: The Cloud Feedback Model Intercomparison Project (CFMIP) contribution to CMIP6, Geosci. Model Dev., 10, 359-384, doi:10.5194/gmd-10-359-2017, 2017. Corrected in text.